# SaltISNet3D: Interactive Salt Segmentation from 3D Seismic Images Using Deep Learning

Hao Zhang *, Peimin Zhu and Zhiying Liao

School of Geophysics and Geomatics, China University of Geosciences, Wuhan 430074, China
* Correspondence: zhang_hao_igg@cug.edu.cn

**Abstract:** Salt interpretation using seismic data is essential for structural interpretation and oil and gas exploration. Although deep learning has made great progress in automatic salt image segmentation, it is often difficult to obtain satisfactory results in complex situations. Thus, interactive segmentation with human intervention can effectively replace the fully automatic method. However, the current interactive segmentation cannot be directly applied to 3D seismic data and requires a lot of human interaction. Because it is difficult to collect 3D seismic data containing salt, we propose a workflow to simulate salt data and use a large amount of 3D synthetic salt data for training and testing. We use a 3D U-net model with skip connections to improve the accuracy and efficiency of salt interpretation. This model takes 3D seismic data volume with a specific size as an input and generates a salt probability volume of the same size as an output. To obtain more detailed salt results, we utilize a 3D graph-cut to ameliorate the results predicted by the 3D U-net model. The experimental results indicate that our method can achieve more efficient and accurate segmentation of 3D salt bodies than fully automatic methods.

**Keywords:** 3D U-net; 3D graph-cut; salt interpretation; interactive segmentation method





## 1. Introduction

As a current popular development direction, artificial intelligence has not only gradually improved diverse application scenarios but also plays an indispensable role in the intelligent transformation and upgrading of various industries. Against the background of the current big data era, deep learning, as an important branch of artificial intelligence, can mine and analyze valuable information and important characteristics from data, which has become a widely recognized intelligent research strategy [1]. At present, many researchers have begun to introduce deep learning technology for interdisciplinary research. It has been widely used in natural language processing [2–4], image recognition and classification [5–7], assisted medical diagnosis [8], autonomous driving [9], earth science [10,11], etc. In recent years, deep learning has made significant progress in the field of seismic exploration, such as first-break picking [12,13], fault detection [14–17], horizon tracking [18,19], velocity analysis [20,21], and salt identification [22]. The work of this paper focuses on identifying salt in 3D seismic data.

A salt dome is a mushroom-shaped geological structure that exists underground and has good airtightness, which provides the basic conditions for oil and gas accumulation and storage [23]. Therefore, the accurate interpretation of salt is of great significance for the exploration and development of subsalt reservoirs. However, salt interpretation based on 3D seismic data is still challenging. The complex shape, the large dip angle of the stratum, and the higher velocity of the salt compared with the surrounding rock result in a complex seismic reflection signal, and structural distortion in the time domain [24]. In addition, with the improvement in the exploration of underground salt, the amount of seismic data is increasing, which leads to the problems of long periods, great difficulty, and multiple

solutions of interpretation. Traditional multi-attribute methods have struggled to fulfil the requirements of salt interpretation.

To solve these problems, many computer-assisted interpretation techniques have been proposed. Shi et al. use an image-segmentation method to solve the global optimization problem to detect salt [25], but the calculation cost is high, and it is not suitable for real-time seismic interpretation [26]. Zhou et al. use edge-detection technology to identify salt boundaries, which is simple and efficient [27]. Aqrawi et al. use edge-detection technology to better detect salt [28]. However, the salt recognition method based on edge detection can only achieve good results when the seismic amplitude changes sharply, and the use of seismic amplitude information alone cannot fully reflect a salt feature. Seismic attributes can more intuitively reflect special geological structures and are widely used for salt interpretation. Berthelot et al. propose a salt detection method based on three texture attributes: gray-level co-occurrence matrix (GLCM) attributes, frequency-based attributes, and dip and similarity attributes. They demonstrate that the classification performance improves by combining at least two texture attributes [29]. Shafiq et al. use a texture gradient to measure the texture difference between two adjacent time windows to detect salt boundaries [30]. In addition, seismic attributes such as the gradient structure tensor [31], discontinuities [32], and salt likelihoods [33] have been successfully applied to salt interpretation and have achieved certain results. However, interpretation methods based on seismic attributes usually need to extract multiple attributes and use special processing methods to extract seismic information. They then perform fine interpretations to obtain the final results, which require a lot of effort and time.

Deep learning has rapidly developed and gradually has been applied to seismic interpretation, in which a research boom has been set off in the field of salt identification. Waldeland and Solberg [34] use a convolution neural network (CNN) to predict the salt in 2D seismic profiles. Shi et al. [35] propose a 3D CNN model to achieve salt interpretation in 3D seismic data that can capture salt features without manual input. Guo et al. [36] propose a supervised deep learning method for effectively segmenting salt bodies. The method designs an edge-prediction branch to predict salt boundaries, which guides feature learning by supervising the loss function, thus distinguishing the features on both sides of the salt boundaries. However, the salt identified by fully automatic methods based on deep learning is still not perfect. Therefore, it is worthwhile to use interactive segmentation methods to refine the results. There are few related studies on interactive seismic image segmentation using deep learning. Shi et al. [37] utilize a flood-filling network to interactively extract salt bodies. The network takes, as the combined input, the previous mask output along with a seismic image in a new moving window to predict the salt in this window. Zhang et al. [38] propose combining a CNN and a graph-cut to interactively extract salt in 2D seismic profiles. However, these methods still have the following problems: (1) Due to the small sample size of the training set, the results predicted by the pre-trained models are defective, which affects the final recognition result. In particular, when these methods are applied to new data, more user interactions are still required to achieve satisfactory results. (2) These methods are mainly aimed at the problem of automatic salt identification in 2D seismic profiles, so the processing efficiency of 3D seismic data is inadequate.

To address the problem of insufficient samples, we present a method for randomly generating 3D seismic data containing salt, which aims to leverage large amounts of synthetic data to train CNN models. In addition, to improve the efficiency of salt interpretation, we utilize a 3D U-net model with skip connections to predict the salt probability from seismic data, and then use the 3D graph-cut with improved edge-weights to obtain more detailed 3D salt results. Experiments indicate that our method is not only superior to the fully automatic CNN method; it can also use less interactions to obtain more accurate segmentation results than the 2D interactive segmentation method.

## 2. Methods

Using deep learning technology to improve the accuracy of automatic salt segmentation usually requires huge seismic datasets to train and verify a CNN model. However, 3D seismic data containing salt are difficult to collect, and manual salt interpretation can be time-consuming and subjective. To avoid these problems, we present a method to rapidly and randomly simulate a large amount of seismic data containing salt structures for training and validating CNN models.

### 2.1. Synthetic Salt Data

Figure 1 shows a flow chart of 3D seismic data simulation with salt. This scheme can add various shapes of salt to 3D seismic data to meet the requirements of the CNN model for data diversity. The basic steps for modeling 3D seismic data containing salt are as follows:

(1) Constructing a random polyhedron: Figure 1a shows a polyhedron used to construct a 3D salt body. A polyhedron is formed by connecting 10 to 30 random points, making them random and diverse to meet various salt structures.

(2) Building binary label data of a 3D salt body: we use the cubic spline interpolation algorithm to smooth the polyhedron from step (1) and fill the smoothed polyhedron in a data volume with a size of $128 \times 128 \times 128$ to construct a binary salt label, as shown in Figure 1b.

(3) Building a layered acoustic impedance model: Figure 1c shows a layered geological model that is composed of the wave impedance of sandstone and mudstone. The acoustic impedance is obtained by multiplying the velocity and density of rock. The velocity and density of the sandstone and mudstone in the model are mainly based on the empirical formula proposed by Gardner [39]. Here, we set the range of the P-wave velocity of sandstone ($V_s$) to 1.5–2.5 km/s and calculate the density of sandstone ($\rho_s$) using the formula $\rho_s = 1.5695 V_s^{0.3025}$. The range of the P-wave velocity of mudstone ($V_m$) is 2.7–4.3 km/s and the formula of density for the mudstone ($\rho_m$) is $\rho_m = 1.6385 V_m^{0.2924}$.

(4) Building a folded acoustic impedance model: the folded model in Figure 1d is formed by adding a shift field $L_1$ and folded field $L_2$ [40,41] to the layered model in Figure 1c. This method is mainly referred to by Wu et al. [42] for building folded structure models when simulating fault data. The shift field of the inclined structure is defined as:

$$L_1(x, y, z) = ax + by + c \tag{1}$$

where $a$ and $b$ are the slopes in the $x$ and $y$ directions, respectively, the value range is $[-0.25, 0.25]$, $c$ is the intercept, and the value range is $[-20, 20]$.

The folded field $L_2$ is composed of a depth-weighted function and several Gaussian functions:

$$L_2(x, y, z) = \frac{1.5}{z_{\max}} z \sum_{k=1}^{N} b_k e^{\frac{(x-c_k)^2+(y-d_k)^2}{2\sigma_k^2}} \tag{2}$$

where $N$ is the number of Gaussian functions; $(c_k, d_k)$, $\sigma_k$ and $b_k$ represent the center position, width and amplitude of the $k-$th Gaussian function, respectively; $\sigma_k$ and $b_k$ mainly control the range and magnitude of the folded field, respectively; and $1.5/z_{\max} \cdot z$ is a linear scalar function whose main purpose is to make the bending degree of the folds gradually decrease with the increase in depth. Therefore, in order to be closer to the formation of the field seismic data, we can simulate different degrees of folded fields by superimposing several Gaussian functions.

(5) Building a 3D acoustic impedance model containing salt bodies: combined with the binary salt label obtained in step (2), we fill the part of the salt corresponding to the folded model in step (4) as the acoustic impedance value of the salt. Here, we

set the P-wave velocity and the density range of the salt dome to 4.5–6.5 km/s and 2.15–2.44 g/cm$^3$, respectively. A 3D folded model with salt is shown in Figure 1e.

(6) Generating 3D seismic data: we calculate the reflection coefficient model from the 3D folded model containing salt obtained in step (5) and convolve it with the Ricker wave to generate 3D seismic data (Figure 1f).

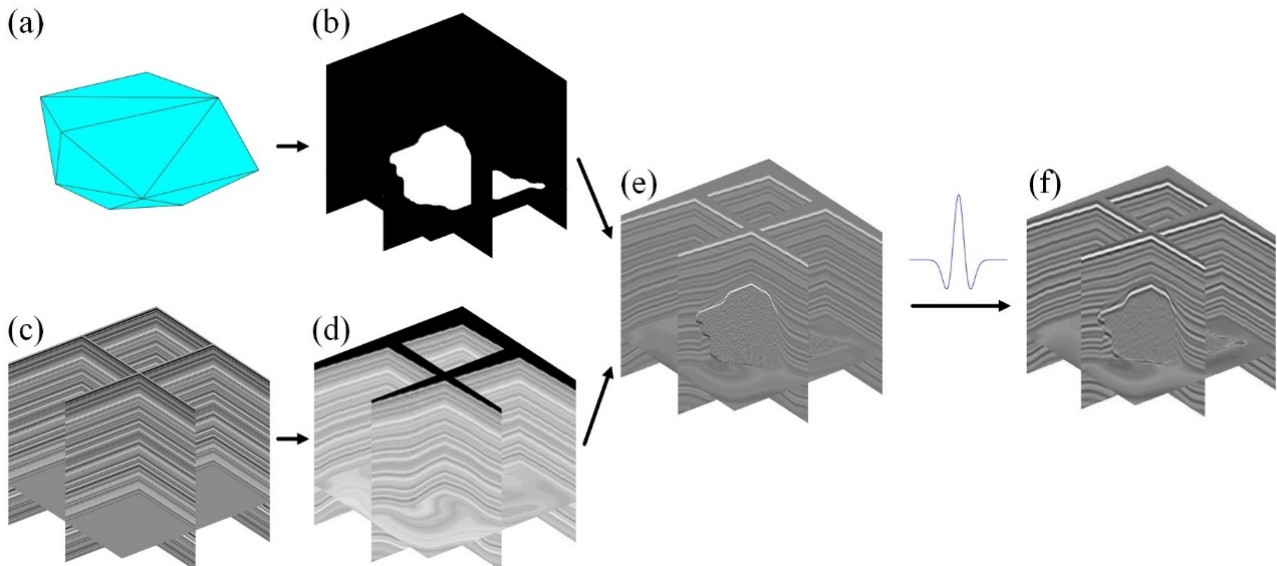

**Figure 1.** A simulation flow chart of 3D seismic data with salt bodies: (**a**) polyhedral; (**b**) a binary label of 3D salt; (**c**) a layered model; (**d**) a folded model; (**e**) a folded model with salt bodies; and (**f**) 3D seismic data with salt bodies.

We simulate 3D seismic data containing salt by convolving the reflection coefficient model with the Ricker wave. This method can synthesize a large amount of 3D seismic data and corresponding salt label data and the generated folded structures and salt structures are diverse and have the characteristics of the contact relationship between salt and the stratum. Figure 2 shows synthetic 3D seismic data and its salt label, the size of which is $128 \times 128 \times 128$. In the salt label, the area representing the salt is assigned a value of 1, and the area that is not salt is assigned a value of 0.

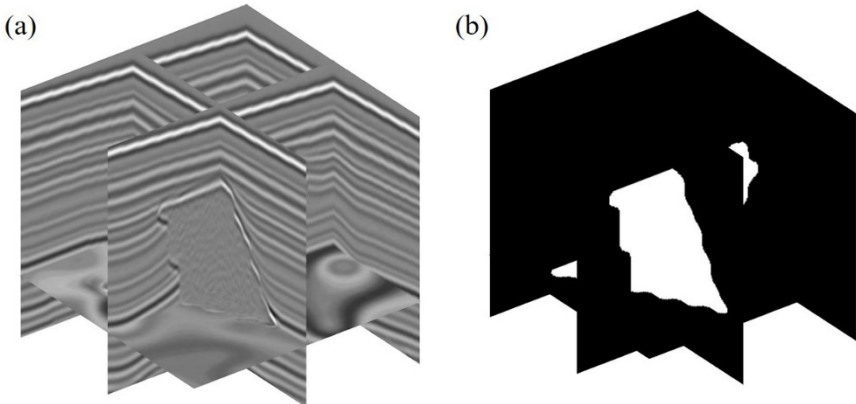

**Figure 2.** Three-dimensional seismic data (**a**) and corresponding salt label (**b**).

### 2.2. Three Dimensional Euclidean Distance Maps

In order to apply the user's interactions to salt recognition, we refer to the 2D interaction scheme in [38] to convert the interaction points into 3D Euclidean distance maps (3D

EDMs). Assuming that the coordinates of the interaction point in the 3D space are $(l, m, n)$, the 3D EDM of the point is calculated as

$$E_{l,m,n} = \begin{cases} e^{-\sigma[(i-l)^2+(j-m)^2+(k-n)^2]}, & \sqrt{(i-l)^2+(j-m)^2+(k-n)^2} < L \\ 0 & , \sqrt{(i-l)^2+(j-m)^2+(k-n)^2} \geq L \end{cases} \quad (3)$$

where $(i, j, k)$ represent the coordinates of each voxel in the 3D data; $\sigma$ is the expansion coefficient, which can be used to control the affected area of the interaction points; and $L$ is the cutoff distance, which can determine the maximum affected area of the interaction points. Figure 3 shows the conversion process of the 3D Euclidean distance maps. Figure 3a shows the 3D seismic data and interaction points, where the red points used to guide the salt are positive interaction points and the blue points used to guide the background are negative interaction points. Figure 3b,c represent 3D Euclidean distance maps of positive and negative interaction point transitions, respectively. The range of values in the two 3D Euclidean distance maps is $[0, 1]$, and it decreases with the distance from the interaction point until it reaches 0.

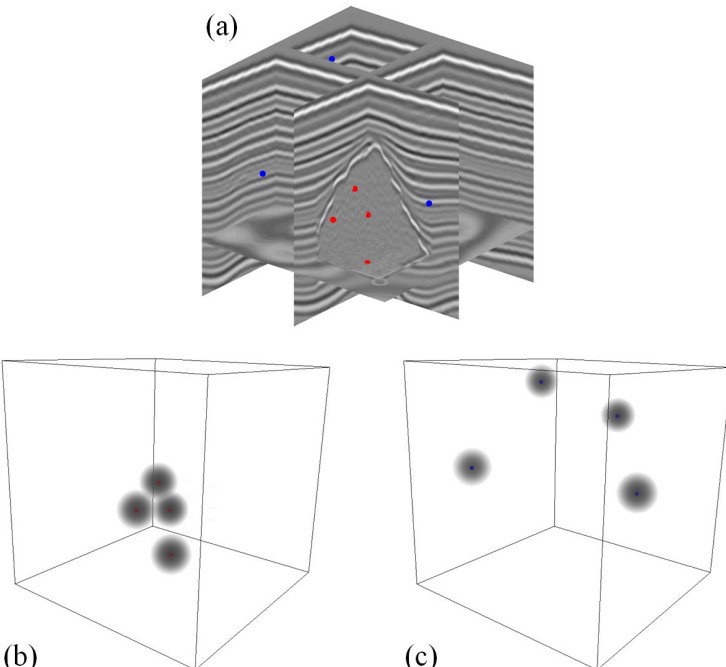

**Figure 3.** Three-dimensional seismic data and the Euclidean distance maps of the interaction points. (**a**) Three-dimensional seismic data and the interaction points; (**b**) the Euclidean distance maps of the positive points; and (**c**) the Euclidean distance maps of the negative points.

### 2.3. Three-Dimensional Network Architecture

We use a 3D U-net architecture to construct a CNN model for 3D salt identification, as shown in Figure 4. In order to make the model suitable for 3D data, we use 3D operators to replace the convolutional layer, pooling layer, and upsampling layer in the 2D U-net model [38]. Since there is a positive correlation between the receptive field in the 3D U-net model and the size of the input data, a larger receptive field can enable the model to extract global information from seismic data. In addition, if the 3D data are divided into small-volume data as an input, the prediction of adjacent small-volume data may cause a duplication or error, so additional processing is still required to improve the prediction results. However, larger receptive fields require more computing resources without reducing the network complexity and calculations. Therefore, the size of the input data or receptive fields is limited by hardware devices.

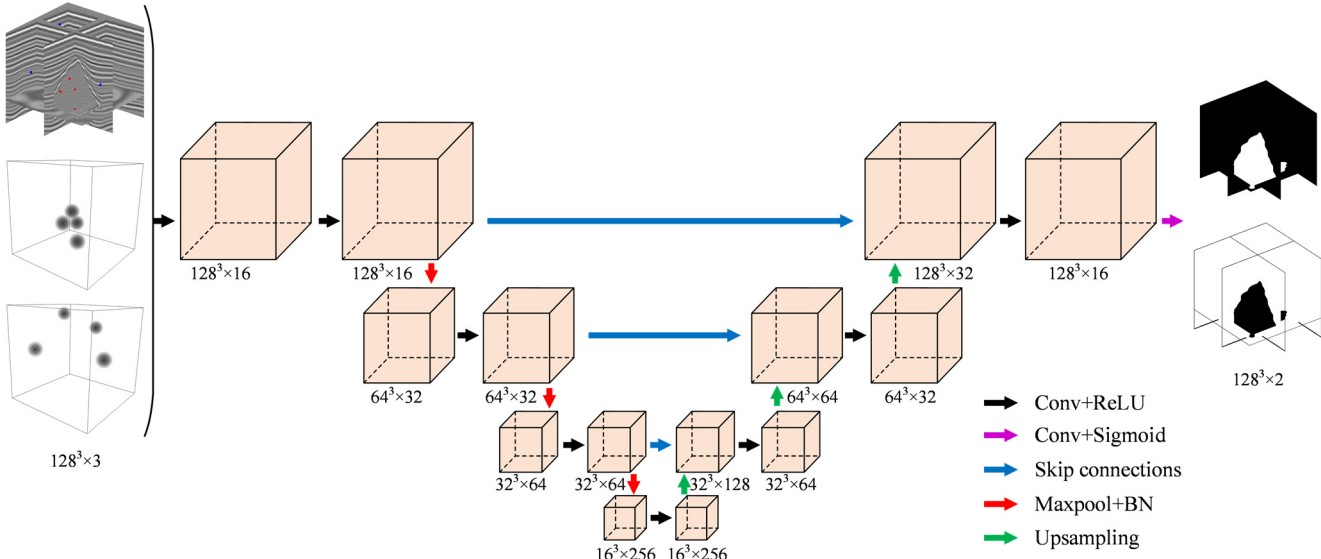

**Figure 4.** The 3D U-net model for 3D salt detection. Conv + ReLU and Conv + Sigmoid represent the addition of the ReLU function and the Sigmoid function after the convolutional layer, respectively. Skip connections indicate that the output of each group in the encoder is used as the decoder input. Maxpool + BN means that each group is followed by maximum pooling and batch normalization operations.

In Figure 4, we can observe that the input data of the 3D U-net model consists of three parts, namely the 3D seismic data and the 3D Euclidean distance map of two interaction points, and their size is set to $128 \times 128 \times 128$. The 3D U-net model consists of an encoder and a decoder. The encoder contains four groups of convolutional layers, and each group of convolutions is connected by a maximum pooling layer. The size of the pooling kernel is $2 \times 2 \times 2$. The sizes of the output feature maps of each group of convolutional layers are $128^3$, $64^3$, $32^3$, and $16^3$, and the numbers of feature maps are set to 16, 32, 64, and 256. The kernel size of all convolutional layers is $3 \times 3 \times 3$, and after each convolutional layer, an activation function is added and a batch normalization operation is performed [43]. The activation function is the ReLU function, and its expression is $Relu(x) = \max\{0, x\}$. Another part that is symmetrical to the encoder is the decoder, which also includes four groups of convolutional layers, and each group of convolutional layers is upsampled by a multiple of $2 \times 2 \times 2$. The input of each convolutional group consists of the output of the corresponding encoder and the upsampled layer. At the end of the model, the final convolution layer, we set the sigmoid function to output the probability of the salt and the background. Due to the symmetry of the U-net architecture, the size of the output data is consistent with the input and it is also $128 \times 128 \times 128$. Compared with the original 2D U-net, the 3D model for salt identification in 3D seismic data has more parameters. In order to give this model a strong generalization ability, we add some dropout layers [44] to the 3D U-net model. To extract salt from 3D seismic data, the cross-entropy loss function is applied to train the model, which is defined as

$$Loss = -\frac{1}{m}\sum_{i=1}^{m}\left(y_i \log y_i' + (1 - y_i)\log(1 - y_i')\right) \qquad (4)$$

where $m$ is the number of samples, $y_i$ is the ground truth, and $y_i'$ is the result predicted by our model.

### 2.4. Three-Dimensional Graph-Cut

Graph-cut is a popular energy optimization algorithm that has been widely used in medical image segmentation [45,46], hyperspectral image classification [47,48], SAR image

classification [49,50], multi-view clustering [51–53], and so on. Because there are still some noise and wrong predictions in the 3D salt results predicted by the 3D U-net model, we utilize a graph-cut with improved edge weights to obtain more refined results. The success of this algorithm mainly depends on the correct identification of the boundary between the salt and the background. Therefore, constructing a suitable energy function for the graph-cut algorithm is one of the important factors for identifying the salt bodies in 3D seismic data.

We construct a graph based on 3D seismic data, which is defined as a 3D weighted undirected graph. $G = G(V, E)$, where $V$ represents the set of nodes in the graph and $E$ represents the set of all edges in the graph. We assume that $X$ represents all voxels in the 3D data, and each voxel $x \in X$ is represented as a node of the graph $G$. Furthermore, the graph $G$ has two special terminal nodes, namely the source $S$ and the sink $T$, as shown in Figure 5. There are two types of edges in the graph, namely t-links and n-links. The t-links represent the edges connecting each node $x$ to the source $S$ and the sink $T$. $Ne(x)$ denotes the neighborhood set of $x$, and the n-links represent the edges between each node $x$ and its adjacent node $y$, where $y \in Ne(x)$.

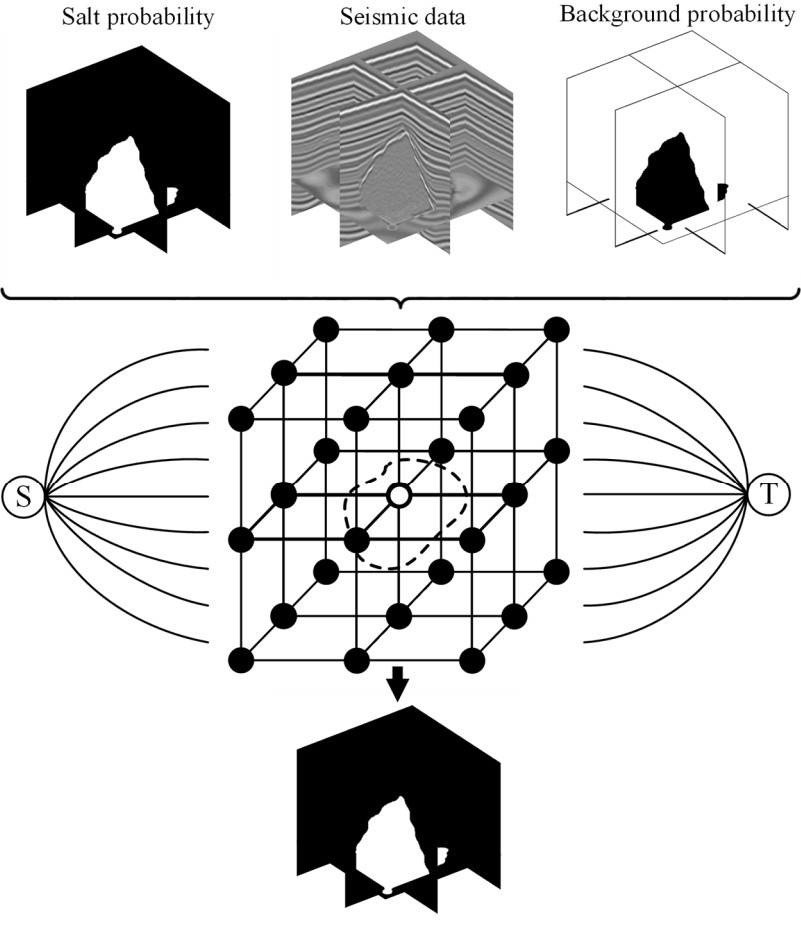

**Figure 5.** Flow chart of salt segmentation based on 3D graph-cut. The input of the 3D graph-cut includes the salt dome probability, background probability, and seismic data. $S$ and $T$ represent the salt terminal and the background terminal in the 3D graph, respectively. The dotted line represents the optimal segmentation of the 3D graph.

For the problem of 3D image segmentation, we present an improved graph-cut algorithm based on edge weights, assuming that $A$ is defined as a segmentation; that is,

all voxels are classified into salt and background. According to the method proposed by Boykov and Jolly [54], the energy function $C(A)$ can be defined as

$$C(A) = \lambda \cdot R(A) + B(A) \tag{5}$$

where $R(A)$ and $B(A)$, respectively, represent the regional term and the boundary term and $\lambda$ is an importance factor that determines their relative importance. The mathematical expressions of the two terms are as follows:

$$R(A) = \sum_{x \in X} R_x(A_x) \tag{6}$$

$$B(A) = \sum_{x \in X, y \in Ne(x)} B_{\{x,y\}} \tag{7}$$

Based on the energy function $C(A)$, we can find that (1) if a voxel is located inside the salt or background, the regional term becomes more important. In this case, voxels are usually surrounded by voxels of the same category, so their probability of belonging to this category is significantly higher than that of belonging to other categories. (2) If a voxel is located near the boundary between the salt and the background, the boundary term $B(A)$ is more important. Therefore, we modify the regional term and the boundary term to enhance the contrast between the salt bodies and their surrounding background in the 3D seismic data. $O$, $B$, and $I_x$ are voxels belonging to the salt, voxels belonging to the background, and the pixel value of voxel $x$, respectively. In addition, $\Pr(I_x|O)$ and $\Pr(I_x|B)$, respectively, represent the probabilities that a voxel belongs to the salt and background, which can be obtained from the output of the model. $K(x, y)$ is the probability that two adjacent voxels $x \in X$ and $y \in Ne(x)$ belong to the same class, which is calculated using $\Pr(I_x|O)$, $\Pr(I_x|B)$, $\Pr(I_y|O)$, and $\Pr(I_y|B)$. The improved boundary term is given below:

$$B_{\{x,y\}} = K(x,y) \cdot exp(-\frac{(I_x - I_y)^2}{2\varepsilon^2}) \cdot \frac{1}{d(x,y)} \tag{8}$$

where $d(x, y)$ represents the distance between two voxels, $x$ and $y$.

The sub-items $R_x(salt)$ and $R_x(background)$ of the regional term are defined as:

$$R_x(salt) = -K_x^{salt} \exp(-((S_A - S_B)/D)^2) \ln(\Pr(I_x|O)) \tag{9}$$

$$R_x(background) = -K_x^{bg} \exp(-((S_A - S_B)/D)^2) \ln(\Pr(I_x|B)) \tag{10}$$

where $K_x^{salt}$ and $K_x^{bg}$ rely on the probabilities that a voxel $x$ belongs to the salt body and the background, respectively. They are calculated by comparing the $\Pr(I_x|O)$ and $\Pr(I_x|B)$. If the probability of $x$ belonging to the salt body is higher, $\Pr(I_x|O)$ is assigned to $K_x^{salt}$ and $\Pr(I_x|B)$ is assigned to $K_x^{bg}$. Otherwise, $\Pr(I_x|O)$ is assigned to $K_x^{bg}$ and $\Pr(I_x|B)$ is assigned to $K_x^{salt}$. $K(x,y)$, $K_x^{salt}$, and $K_x^{bg}$ can be used to deal with the problem of low contrast between salt bodies and the background. The $S_A$, $S_B$, and $D$ in Equations (9) and (10) reduce the sensitivity of the parameter $\lambda$ to the graph-cut algorithm. Their expressions are defined as

$$S_A = \sum_{i=1}^{|A_y|} \Pr(I_y|O) / d(x, y_i) \tag{11}$$

$$S_B = \sum_{i=1}^{|B_y|} \Pr(I_y|B) / d(x, y_i) \tag{12}$$

$$D = \sum_{i=1}^{|Ne(x)|} (1/d(x,y_i)) \tag{13}$$

where $A_y$ and $B_y$ are the neighborhood sets of $x$ in the cases of $\Pr(I_y|O) > \Pr(I_y|B)$ and $\Pr(I_y|O) < \Pr(I_y|B)$, respectively; $S_A$ represents the total distance weighting probability of adjacent voxels belonging to the salt in the set $A_y$; $S_B$ represents the total distance weighting probability of adjacent voxels belonging to the background in the set $B_y$; and $D$ is a normalization factor. From Equations (11) and (12), if a voxel is located inside the salt area, we can conclude that $S_A \gg S_B$. Otherwise, if a voxel is located inside the background area, we can conclude that $S_A \ll S_B$. In both cases, the exponential terms in Equations (9) and (10) have smaller values compared with the exponential term in Equation (8), which reduces the cost of the regional term and gives the boundary term a higher proportion. If a voxel is located at or near the boundary between the salt body and the background, we can conclude that $(S_A - S_B)^2$ will tend to 0. The exponential terms in Equations (9) and (10) have larger values, which makes the regional term more important. Therefore, this improvement uses the position of a voxel to obtain the probability of belonging to a certain class and corrects the cost of the regional item, thereby providing an appropriate weight between the regional item and the boundary item.

Figure 5 shows a flow chart of the salt segmentation based on 3D graph-cut. We use the 3D seismic data and the salt probability and background probability predicted by the 3D U-net model as the input of the 3D graph-cut, where the salt probability is used as the $\Pr(I_x|O)$ of Equation (7) and the background probability is used as the $\Pr(I_x|B)$ of Equation (8). The 3D graph is composed of nodes, edges, the salt terminal $S$, and the background terminal $T$. The number of nodes is the same as the number of voxels in the 3D seismic data. According to the steps of the 3D graph-cut algorithm introduced above, we calculate $K(x,y)$, $K_x^{salt}$, $K_x^{bg}$, and the costs of the regional item and the boundary item in turn, and then use the max-flow or min-cut [55–57] to find the optimal value of the energy function $C(A)$, thereby obtaining a better salt segmentation result (shown by the dotted line in Figure 5). The complete calculation process of the 3D graph-cut is shown in Algorithm 1.

---

**Algorithm 1** Three-dimensional graph-cut

---

**Input:** Salt probability $\Pr(I_x|O)$, background probability $\Pr(I_x|B)$ and seismic data $I$
**Output:** salt segmentation $A$
**for** $x$ in $X$ **do**
    Obtain the probabilities of adjacent voxels $\Pr(I_y|O)$ and $\Pr(I_y|B)$, where $y \in Ne(x)$.
    Obtain $K(x,y)$ using $\Pr(I_x|O)$, $\Pr(I_x|B)$, $\Pr(I_y|O)$, $\Pr(I_y|B)$
    Obtain the cost of the boundary term $B(A)$ from $B_{\{x,y\}}$ by Equation (8)
    Obtain $K_x^{salt}$, $K_x^{bg}$ using $\Pr(I_x|O)$ and $\Pr(I_x|B)$
    Obtain $S_A$, $S_B D$ by Equation (11), Equation (12) and Equation (13), respectively.
**end for**
    Obtain the cost of the regional term $R(A)$ from $R_x(salt)$ by Equation (9) and
    $R_x(background)$ by Equation (10)
    Obtain the energy function $C(A)$ by Equation (5)
    Apply max-flow/min-cut to minimize the energy function $C(A)$
**Return** salt segmentation $A$

---

## 3. Experiments

### 3.1. Training Using the Synthetic Salt Data

To train and test the model, we adopt the 3D salt data simulation method introduced in Section 2.1 to synthesize a lot of seismic data samples. This dataset contains a total of 1000 pairs of 3D seismic data and corresponding salt labels. The size of each seismic data sample was $128 \times 128 \times 128$. There are two classes of salt labels: salt (white) and

background (black). To expand the number of samples, we perform data augmentation on this synthetic dataset, including methods such as rotation, flipping, and mirroring. At the same time, we normalize all input data to ensure stability during training. During training, we divide the training dataset into small batches and input them into the network model. A larger batch size can make the learning process more stable. However, due to the limitations of computing resources, the batch size cannot be set too large. For example, the GPU (NVIDIA Tesla V100S) used in this experiment has 32 GB, which limits the batch size to 5. In addition, in order to avoid overfitting of the model, we randomly select a batch from the training set for training.

In order to verify that the 3D EDMs and 3D graph-cut algorithm can effectively improve the accuracy of salt identification, we use the 3DU-net, 3DU-net-EDMs and 3DU-net-EDMs-GC methods on the synthetic 3D salt data. The 3DU-net method is a fully automatic salt identification method based on deep learning, and its input data are only 3D seismic data, while the 3DU-net-EDMs and 3DU-net-EDMs-GC methods require 3D seismic data and two types of 3D Euclidian distance maps. Since there are no user interaction points, we randomly generate interaction points whose number range is set to [1,20]. The 3DU-net-EDMs-GC method combines 3DU-net, 3D EDMs, and the 3D graph-cut algorithm, and its CNN model is consistent with 3DU-net-EDMs. Therefore, we keep the parameters of the 3DU-net-EDMs as the CNN part of the 3DU-net-EDMs-GC method.

To compare the performances of different methods in the case of insufficient samples, we select two different training/testing ratios of 4:1 and 1:4. At the beginning of training, the parameters of all models are initialized based on the method proposed by He et al. [58]. We use the cross-entropy loss function and the adaptive moment estimation (Adam) operator [59] to iteratively update the training parameters of all models. The training process of this experiment includes 100 epochs, and all the training datasets are learned once in each epoch. To effectively improve the convergence speed of the models, the learning rate of the initial training stage is set to $1 \times 10^{-4}$, and the learning rate is adjusted with a discounting factor of 0.9 with the increasing epochs. Figure 6 shows the loss curve of different models at different ratios. In Figure 6a, we can see that when the ratio is 4:1, the loss values of the two models of 3DU-net and 3DU-net-EDMs on the training dataset and testing dataset decrease with the increase in training epochs. After training for 40 epochs, all loss values decrease slowly or even remain unchanged. The 3DU-net and 3DU-net-EDMs models converge to 0.22 and 0.195 on the training dataset, respectively, and converge to 0.24 and 0.22 on the testing dataset. Compared with the training dataset, the loss curves of both models on the testing dataset indicate poorer performance, and the loss values are more unstable, which was mainly caused by the fact that the testing dataset had never been trained. When the ratio is 1:4, the training and testing losses of the 3DU-net-EDMs model decrease with the increase in training epochs and reach a stable state, while the losses of the 3DU-net model fail to converge well, as shown in Figure 6b. It illustrates that the 3DU-net-EDMs model with a 3D Euclidean distance map as an input still has a good performance in the case of insufficient training samples.

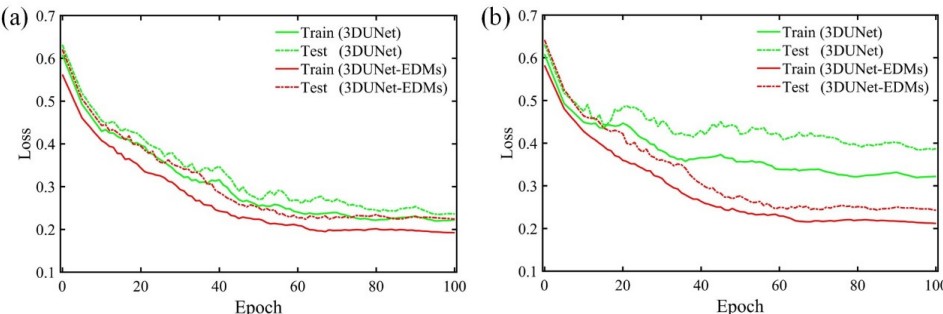

**Figure 6.** The loss curves of different models at two different training/testing ratios of 4:1 (**a**) and 1:4 (**b**).

### 3.2. Test Using Validation Data

We utilize some evaluation metrics on the testing dataset to quantitatively evaluate the performances of the different models. Based on the four indicators of true positive (TP), true negative (TN), false negative (FN), and false positive (FP), we obtain the accuracy, precision, recall, and F1 scores, which can represent the performance of each method. Their expressions are defined as

$$\text{Accuracy} = \frac{\text{TP} + \text{TN}}{\text{TP} + \text{TN} + \text{FP} + \text{FN}} \tag{14}$$

$$\text{Precision} = \frac{\text{TP}}{\text{TP} + \text{FP}} \tag{15}$$

$$\text{Recall} = \frac{\text{TP}}{\text{TP} + \text{FN}} \tag{16}$$

$$\text{F1} = \frac{2 \times \text{Precision} \times \text{Recall}}{\text{Precision} + \text{Recall}} \tag{17}$$

where TP, TN, FN, and FP need to set a threshold for calculation. For example, if the threshold of this experiment is set to 0.5, it means that when the probability of a salt output by the model is greater than 0.5, the voxel is labeled as salt; otherwise, it is labeled as background.

We apply the three methods of 3DU-net, 3DU-net-EDMs, and 3DU-net-EDMs-GC to the testing dataset and count their evaluation metrics using two different training/testing ratios, as shown in Table 1. When the ratio is 1:4, the accuracy, precision, recall, and F1 scores of the 3DU-net-EDMs-GC method are 0.8962, 0.8519, 0.8733 and 0.8607, respectively, which are higher than the corresponding metrics of other methods. Similarly, when the ratio is 4:1, the metrics of the 3DU-net-EDMs-GC method are also higher than those of the other methods, which indicates that the 3D EDMs and the 3D graph-cut algorithm can enhance the accuracy of salt identification. In addition, in the case of insufficient training data (the ratio of 1:4), the metrics of the 3DU-net-EDMs-GC method are better than those of the 3DU-net in the case of sufficient data (the ratio of 4:1). This demonstrates that the 3DU-net-EDMs-GC method can ensure high accuracy of salt identification in the case of insufficient training data.

**Table 1.** Evaluation metrics of different methods at different training/testing ratios.

| Training/Testing | Method | Accuracy | Precision | Recall | F1 |
|---|---|---|---|---|---|
| | 3DU-net | 0.7245 | 0.6578 | 0.7034 | 0.7159 |
| 1:4 | 3DU-net-EDMs | 0.8151 | 0.7483 | 0.7702 | 0.8086 |
| | 3DU-net-EDMs-GC | **0.8962** | **0.8519** | **0.8733** | **0.8607** |
| | 3DU-net | 0.8893 | 0.8435 | 0.8714 | 0.8697 |
| 4:1 | 3DU-net-EDMs | 0.9245 | 0.8847 | 0.9070 | 0.8952 |
| | 3DU-net-EDMs-GC | **0.9706** | **0.9097** | **0.9639** | **0.9448** |

The bold numbers in the table are to show that this method has higher performance in different ratios.

To further verify the effectiveness of the 3DU-net-EDMs-GC method, we select two samples from the testing dataset, as shown in Figure 7. The first, second, and third lines in Figure 7 are the prediction results of 3DU-net, 3DU-net-EDMs, and 3DU-net-EDMs-GC respectively, and the fourth line is the corresponding ground truth. Here, we set a threshold to divide the probability of salt predicted by these methods into binary salt results. In Figure 7, we can find that the results predicted by the 3DU-net-EDMs-GC method are the most similar to the ground truth, while in the results predicted by the fully automatic methods of 3DU-net and 3DU-net-EDMs, based on deep learning, there are varying degrees of noise and errors, and the 3DU-net model performs the worst. This also corresponds to

the index values in Table 1, further indicating that the 3D interactive method (3DU-net-EDMs-GC) combined with deep learning and graph-cuts can extract more accurate 3D salt domes from 3D seismic data.

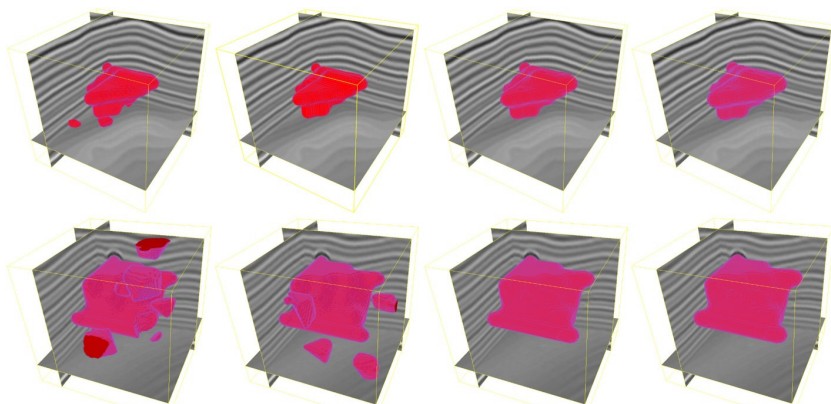

**Figure 7.** The salt segmentation results predicted by different methods. The first, second, third, and fourth columns are 3DUnet, 3DUnet-EDMs, 3DUnet-EDMs-GC, and the ground truth, respectively.

To study the relationship between the accuracy of the 3DU-net-EDMs-GC method in predicting 3D salt bodies and increasing the number of interaction points, we record the performance of the 3DU-net-EDMs-GC method over 20 points on the testing dataset when the training/testing ratio is 4:1. From the curve in Figure 8, we can see that the accuracy of the 3DU-net-EDMs-GC method increases as the number of interaction points increases, and its growth rate initially is fast and then slows down. The accuracy increases quickly from 0.616 to 0.968 when the number of interaction points increases from 1 to 15, but it increases slowly from 0.972 to 0.976 when the number of interaction points increases from 17 to 20. This demonstrates that when the number of interaction points increases to a certain amount, the accuracy improvement brought by an increase in number of interaction points will decrease. Therefore, the 3DU-net-EDMs-GC method needs to balance the relationship between the accuracy rate and the number of interaction points in practical applications. On the basis of ensuring accuracy, we should try to reduce the interactive operations to avoid affecting the efficiency of the seismic interpretation. In addition, Figure 9 shows the qualitative results of the 3DU-net-EDMs-GC method in the cases of different numbers of interaction points. We can observe that the accuracy values of the 3DU-net-EDMs-GC method in predicting salt are 0.715, 0.817, 0.866, and 0.903 when the number of interaction points was 1, 3, 7, and 10, respectively. This also demonstrates that increasing the number of interaction points can improve the accuracy of this method in predicting salt bodies using synthetic seismic data.

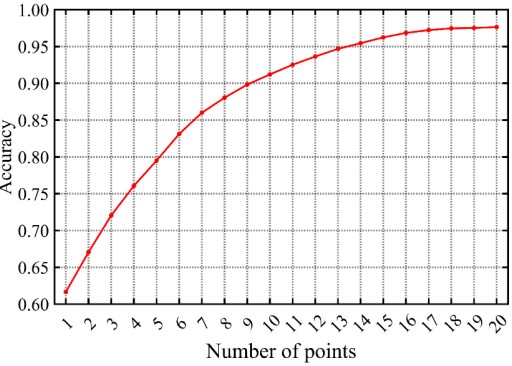

**Figure 8.** The accuracy of the 3DU-net-EDMs-GC method increases with an increasing number of interaction points.

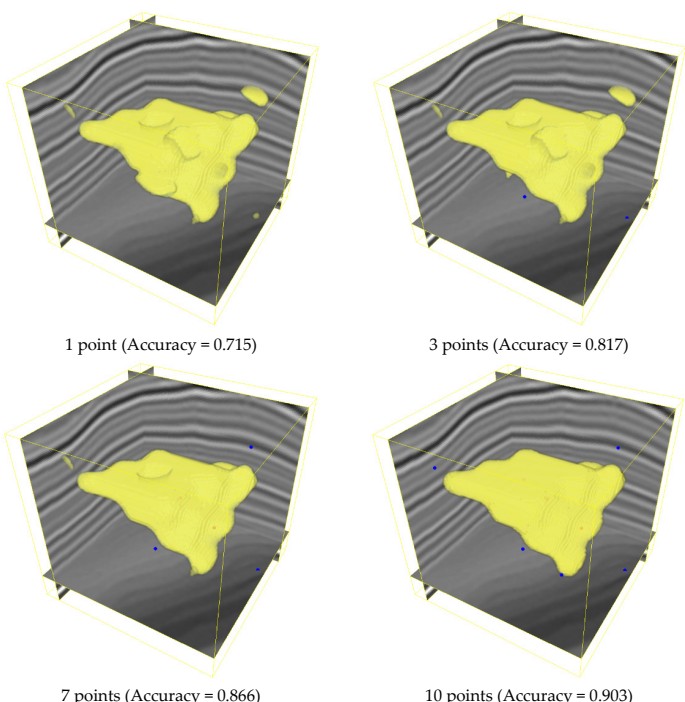

**Figure 9.** The qualitative results of the 3DU-net-EDMs-GC method on the testing dataset. The blue dots represent interaction points.

## 4. Application of 3D Field Seismic Data

### 4.1. SEAM Seismic Data

In order to further verify the performance of the 3DU-net-EDMs-GC method, we utilize it to extract salt bodies from 3D field seismic data. Figure 10 shows the SEAM seismic data, the size of which is $1169 \times 1002 \times 751$, of which the length of the south-north direction (S-N) is 23.38 km, the length of the west-east direction (W-E) is 20 km, and the depth is 15 km. In Figure 10, we can see that the data change strongly at the salt boundaries, which is conducive to the detection of 3D salt bodies. In addition, there are two main salt bodies in the SEAM seismic data, of which one is in the shallow layer with a complex shape, while the other is in the deep layer with a dome shape. Because the input data size of the model is $128 \times 128 \times 128$, the SEAM seismic data are divided into small blocks of the same size. For consistency with the training stage, we similarly normalize the SEAM seismic data.

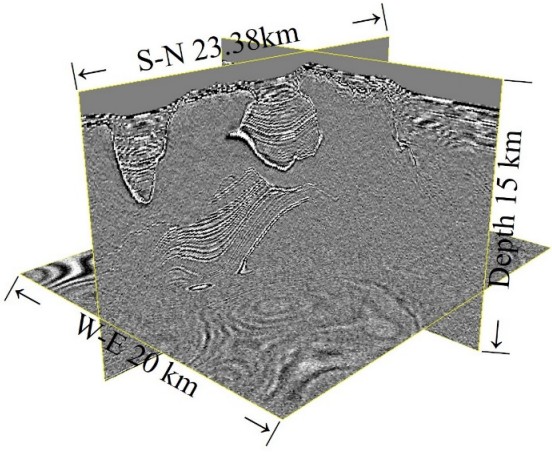

**Figure 10.** SEAM seismic data.

Figures 11–13 indicate the salt segmentation results predicted by the 3DU-net, 3DU-net-EDMs and 3DU-net-EDMs-GC methods using the SEAM seismic data, respectively. (a–c) represent the results predicted by each method on the S-N section, the W-E section, and the horizontal slice, respectively. The blue masks in the figures are the salt results predicted by each method. (d) shows the prediction results of each method in 3D space, which are marked in yellow. We compare the prediction results of the three methods in the E-W section. The salt results predicted by 3DU-net miss some small salt bodies in the left area, and there are some false positives in the right and bottom areas, as shown in Figure 11a. The results shown in Figure 12a are slightly better, and there are only a few false positives on the left area. Compared with the results predicted by the first two methods, the results of the 3DU-net-EDMs-GC method are the most accurate, as shown in Figure 13a. Moreover, we can obtain similar results in the W-E section (Figures 11b, 12b and 13b) and horizontal slice (Figures 11c, 12c and 13c). Comparing the results of the three methods in 3D space, we can observe that Figures 11d and 12d show scattered noise and incorrect salt prediction results. The salt results of the 3DU-net method are the worst, where the salt results predicted by the 3DU-net-EDMs-GC method (Figure 13d) are more complete and closer to the real salt bodies. This proves that the fully automatic salt segmentation method based on deep learning (3DU-net) still has some shortcomings, and its predicted results often have some scattered noise and missing salt bodies. Although the 3DU-net-EDMs method incorporates human interaction information, which can improve the prediction results to a certain extent, it still cannot achieve satisfactory results. The 3DU-net-EDMs-GC method combines human interactive information and the 3D graph-cut, which can not only effectively remove scattered noise or false salt bodies but also supplements undetected salt bodies by adding interactive points, thus improving the accuracy and efficiency of salt body identification in 3D seismic data.

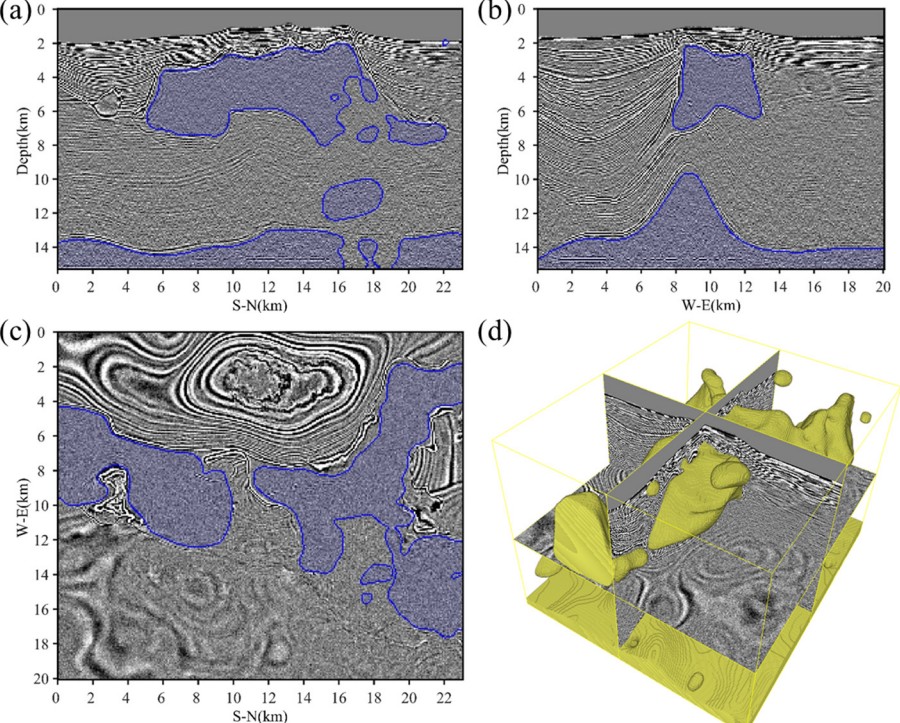

**Figure 11.** The salt results predicted by the 3DU-net method in the SEAM seismic data. (**a**) W-E section; (**b**) S-N section; (**c**) horizontal slice; and (**d**) 3D visualization.

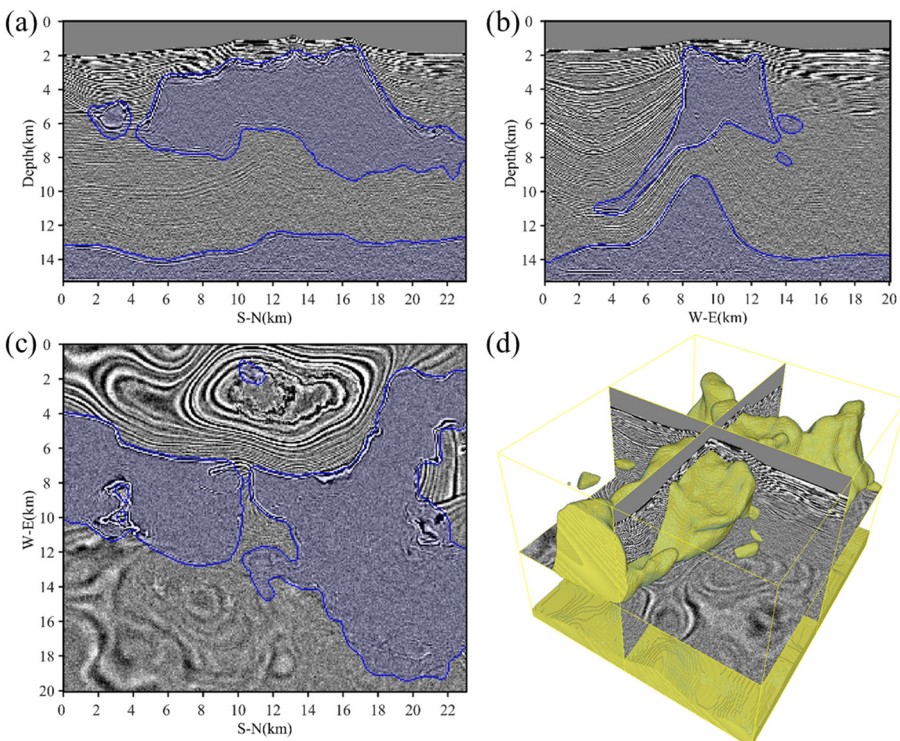

**Figure 12.** The salt results predicted by the 3DU-net-EDMs method in the SEAM seismic data. (**a**) W-E section; (**b**) S-N section; (**c**) horizontal slice; and (**d**) 3D visualization.

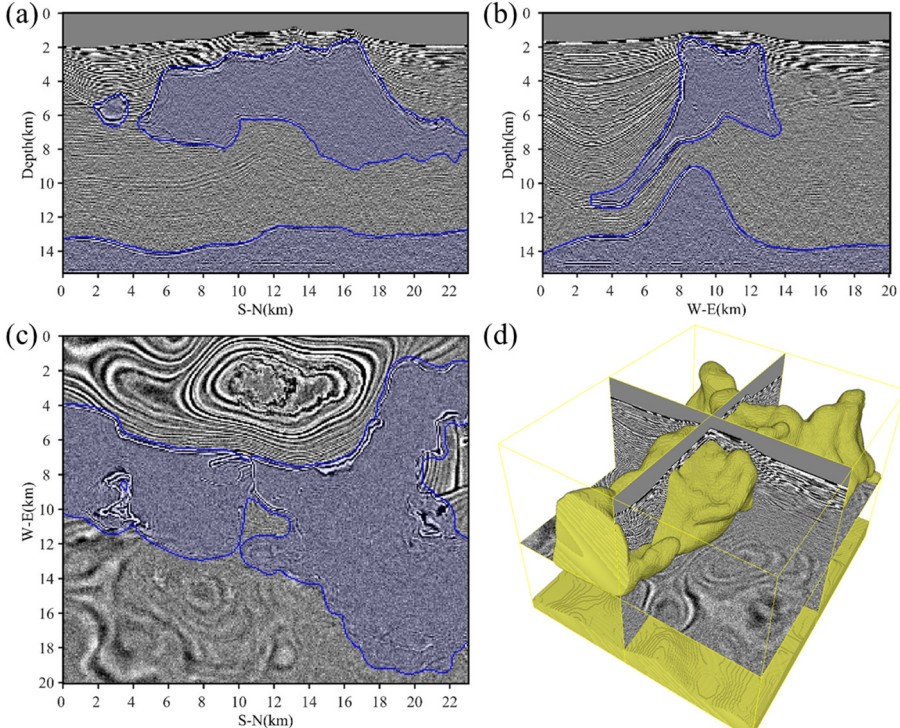

**Figure 13.** The salt results predicted by the 3DU-net-EDMs-CG method in the SEAM seismic data. (**a**) W-E section; (**b**) S-N section; (**c**) horizontal slice; and (**d**) 3D visualization.

*4.2. F3 Block Seismic Data*

We use other field seismic data for testing, which are the Netherlands offshore F3 block seismic data. As shown in Figure 14, we remove the part of the original data that

does not contain salt bodies and crop out a section measuring $500 \times 400 \times 150$. In addition, the red lines in Figure 14 indicate that there are obvious faults in the seismic data, and the yellow arrow indicates the salt bodies in the seismic data.

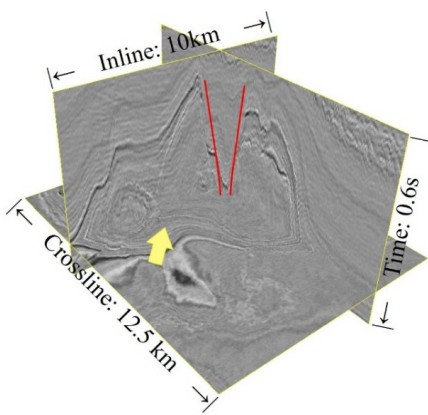

**Figure 14.** The Netherlands offshore F3 block seismic data. The red lines indicate the faults and the yellow arrow indicates the salt bodies in these data.

Figures 15–17 indicate the salt results predicted by the 3DU-net, 3DU-net-EDMs, and 3DU-net-EDMs-CG methods on F3 block seismic data, where (a–d) represent the inline section, crossline section, horizontal slice, and represent the 3D visualization of the salt, respectively. Different from the previous example, the signal-to-noise ratio of these data is poor, there are many faults, and the salt boundaries are not obvious, which brings great interference to the extraction of the salt bodies. In Figures 15–17, we can see that all three methods can predict the main salt bodies, but the specific details of the salt bodies are different. Compared with 3DU-netand 3DU-net-EDMs, the salt results extracted by the 3DU-net-EDMs-CG method are more consistent with the actual situation. This demonstrates that the 3DU-net-EDMs-CG method has good versatility in the application of extracting 3D salt bodies from field seismic data.

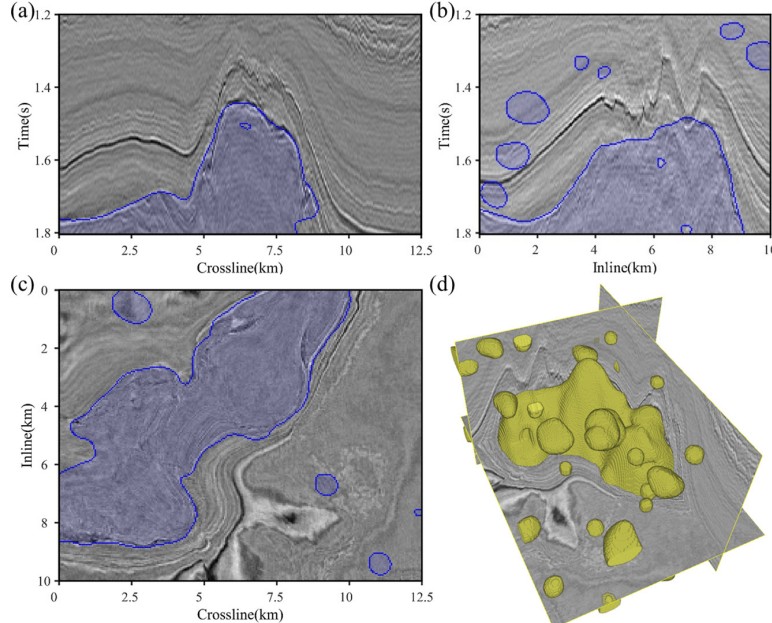

**Figure 15.** The salt results predicted by the 3DU-net method has a number of errors and omissions. in the F3 block seismic data. (**a**) inline section; (**b**) crossline section; (**c**) horizontal slice; and (**d**) 3D visualization.

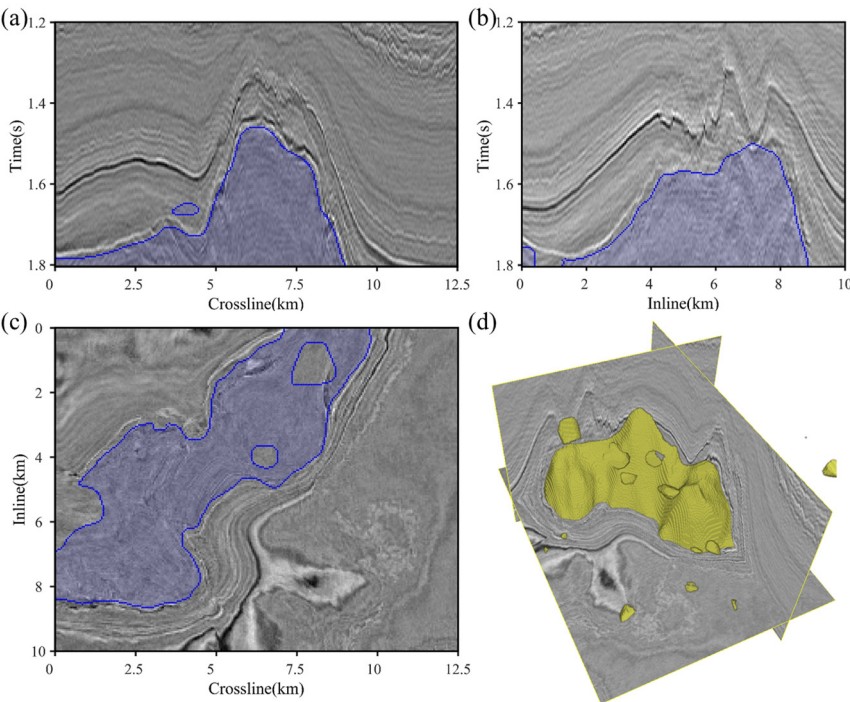

**Figure 16.** The salt results predicted by the 3DU-net-EDMs method has a few small errors in the F3 block seismic data. (**a**) inline section; (**b**) crossline section; (**c**) horizontal slice; and (**d**) 3D visualization.

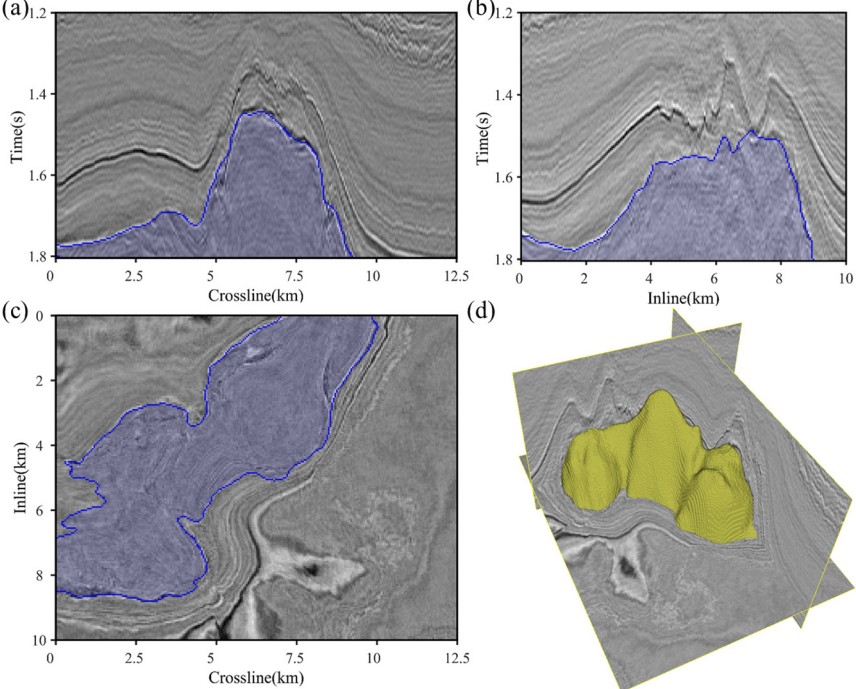

**Figure 17.** The salt results predicted by the 3DU-net-EDMs-CG method has few errors in the F3 block seismic data. (**a**) inline section; (**b**) crossline section; (**c**) horizontal slice; and (**d**) 3D visualization.

## 5. Discussion

Deep learning often requires a large number of samples and a variety of sample types. Although we have proposed a seismic data simulation method for three-dimensional salt mounds, the simulated data are too singular compared with the actual shape of salt

mounds in nature. Therefore, studying how to enrich the diversity of salt mounds as much as possible is worth considering.

The seismic data in the proposed examples are of high quality, and the salt boundaries are obvious. Therefore, in a future work, our method will be applied to complex seismic data for verification, so that it can promote the development of salt interpretation. In addition, this method still requires seismic interpretation experts to add some interaction points, which are used to guide the algorithm to accurately identify salt bodies using 3D seismic data. Considering the efficiency of interpretation work, it is necessary to study the use of fewer interaction points to obtain more accurate results.

## 6. Conclusions

Aiming at the small amount of 3D seismic data containing salt bodies, we propose a simulation method to generate such data randomly. This simulation method can quickly generate a large amount of 3D seismic data containing salt bodies to increase the diversity of seismic data, thus providing a data basis for our method. In addition, for massive 3D seismic data, we also propose an interactive 3D salt interpretation method based on the 3DU-net model and the 3D graph-cut algorithm, namely 3DU-net-EDMs-GC. The interactive method first converts the user's interaction points into 3DEDMs and combines them with 3D seismic data to train the 3DU-net-EDMs model using the synthetic dataset. Due to the presence of scattered noise and incorrect salt bodies in the results predicted by the 3DU-net-EDMs model, we propose a 3D graph-cut with improved edge weights to optimize the results, which can obtain more detailed salt bodies. The field examples demonstrate that the results predicted by our proposed method are more accurate than those predicted by the fully automatic salt segmentation method based on deep learning.

**Author Contributions:** Conceptualization, H.Z., P.Z. and Z.L.; methodology, H.Z. and P.Z.; software, H.Z.; validation, H.Z. and P.Z.; formal analysis, H.Z., P.Z. and Z.L.; investigation, H.Z.; resources, H.Z.; data curation, H.Z.; writing—original draft preparation, H.Z. and Z.L.; writing—review and editing, H.Z. and P.Z.; visualization, H.Z. and Z.L.; supervision, P.Z. and Z.L.; project administration, P.Z.; funding acquisition, P.Z. All authors have read and agreed to the published version of the manuscript.

**Funding:** This research was funded by the Natural Science Foundation of China (Grant No. 42074074).

**Data Availability Statement:** The data presented in this study are available from the corresponding author.

**Acknowledgments:** We would like to express our gratitude to the editor and reviewers for their valuable comments.

**Conflicts of Interest:** The authors declare no conflict of interest.

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
