# Peer review of "SaltISNet3D: Interactive Salt Segmentation from 3D Seismic Images Using Deep Learning"

_remotesensing, doi:10.3390/rs15092319_

Round 1

Reviewer 1 Report

In this paper, the authors propose a simulation method to randomly generate 3D seismic data containing a large number of salt bodies, increasing the diversity of seismic data. At the same time, an interactive 3D salt interpretation method based on a 3DU-type network model and a 3D graph cutting algorithm is proposed with certain accuracy. However, there are still some issues to be resolved in this paper.

(1) In the introduction part, the advantages and disadvantages of traditional methods and deep learning-based methods in detecting salt are introduced, but the outstanding contributions of the method in this paper are not clearly described. Please add the main contribution of the proposed method and describe it in points.

(2) This paper considers fast stochastic simulation of large amounts of seismic data including salt structures as one of its innovative contributions. In the second chapter, the synthesis process of the simulated data is described, but there is no experiment or description to evaluate the quality of the simulated data.

(3) The paper introduces the salt segmentation algorithm based on 3D graph cutting in Section 2.4. For a clearer expression, it is recommended to use the form of pseudo-code.

(4) There are some related works should be discussed by authors, including "Graph-Collaborated Auto-Encoder Hashing for Multi-view Binary Clustering", "Towards Adaptive Consensus Graph: Multi-view Clustering via Graph Collaboration" and "Kernelized multiview subspace analysis by self-weighted learning"

(5) It is suggested to supplement the experiment on the time complexity of the proposed method.

Author Response

Dear Reviewer 1:

Thank you for taking the time to examine our manuscript entitled “SaltISNet3D: Interactive salt segmentation from 3D seismic images using deep learning”. The comments of the two reviewers have been very helpful and we have attempted to address each one in the revised version of the manuscript. Below please find the reviewers’ comments in black text, followed by our responses in red text. Revised portion are highlighted in yellow in the paper. We hope that you will find that the revised version of the manuscript acceptable for publication in Remote sensing.

Sincerely,

Hao Zhang (on behalf of all co-authors)

email: zhang_hao_igg@cug.edu.cn

Reviewer 2 Report

Dear Authors,

I have reviewed the manuscript by Hao Zhang, Peimin Zhu and Zhiying Liao about “SaltISNet3D: Interactive salt segmentation from 3D seismic images using deep learning” as submitted to Remote Sensing. The study is clearly structured and explained presenting new methods of the field as expressed. However, only a few minor improvements should be done in the manuscript, more about the style rather than about the content in order to accept its publication.

English expressions, grammar and also worth use should be considered. E.g., line 430 change from

In addition, there are two main salt bodies in the SEAM seismic data: one is in the shallow layer with a complex shape; the other is in the deep layer with a dome shape.

Into

In addition, there are two main salt bodies in the SEAM seismic data of which one is in the shallow layer with a complex shape, while the other is in the deep layer with a dome shape.

Such minor but needed changes are throughout the manuscript.

E.g., there is an overuse of the word “show” which may be replaced and improved by yield, demonstrate, indicate, illustrate etc. This occurs with a variety of words throughout the manuscript.

In literature, there are certainly some important references mentioned, but I was surprised to have missed so many key references of German and Iranian studies (studies in Iran and Germany) with so many fundamental seismics in salt diapir research as performed in the last decades. It would be worth not to keep them out demonstrating that the status quo of this area has been fully covered.

A few concrete points to clarify include line 138-39, please explain a bit better why you conclude that you can simulate different degrees of folded field by superimposing several different Gaussian functions. It is not conclusive for the reader. Also, the assumption of line 165 about the use of equation 3 remains unclear. Please explain better or rephrase. Correct or translate line 260. Line 296, there are two dotted lines in Figure 5 instead of one, please correct or explain. Line 312, “so we set the batch size to 5”, please explain. Line 489 (also 492, 494), subtitle of figure 15 (also 16 and 17), please explain again.

Such inconclusive assumptions or affirmations at the first sight are throughout the text. It would be recommended to put yourself in the mind of the reader. What may be very clear for you, as you routinely work in this field and your proposed application may be not that clear for the reader in order to reconstruct you line of thought. It would be better sometimes to add a bit more explanation. Once again, the content of the manuscript is great, but it would need a some better forms to express it.

Summarizing, it’s a great manuscript which will need changes in style but not about its content. Acceptation of the manuscript is a must.

Author Response

Dear Reviewers:

Thank you for taking the time to examine our manuscript entitled “SaltISNet3D: Interactive salt segmentation from 3D seismic images using deep learning”. The comments of the two reviewers have been very helpful and we have attempted to address each one in the revised version of the manuscript. Below please find the reviewers’ comments in black text, followed by our responses in red text. Revised portion are highlighted in yellow in the paper. We hope that you will find that the revised version of the manuscript acceptable for publication in Remote sensing.

Sincerely,

Hao Zhang (on behalf of all co-authors)

email: zhang_hao_igg@cug.edu.cn
